# Phase II single arm open label multicentre clinical trial to evaluate the efficacy and side effects of a combination of gefitinib and methotrexate to treat tubal ectopic pregnancies (GEM II): study protocol

Andrew W Horne,[1] Monika M Skubisz,[2] Ann Doust,[1] W Colin Duncan,[1] Euan Wallace,[3] Hilary O D Critchley,[1] Terrance G Johns,[3] Jane E Norman,[1] Siladitya Bhattacharya,[4] Jill Mollison,[4] Michael Rassmusen,[5] Stephen Tong[2]

For numbered affiliations see end of article.

**Correspondence to**
Dr Andrew Horne;
andrew.horne@ed.ac.uk

## ABSTRACT

**Introduction:** Tubal ectopic pregnancy (tEP) is the most common life-threatening condition in gynaecology. tEPs with pretreatment serum human chorionic gonadotrophin (hCG) levels <1000 IU/L respond well to outpatient medical treatment with intramuscular methotrexate (MTX). TEPs with hCG >1000 IU/L take a significant time to resolve with MTX and require multiple outpatient monitoring visits. Gefitinib is an orally active epidermal growth factor receptor (EGFR) antagonist. In preclinical studies, we found that EP implantation sites express high levels of EGFR and that gefitinib augments MTX-induced regression of pregnancy-like tissue. We performed a phase I toxicity study administering oral gefitinib and intramuscular MTX to 12 women with tEPs. The combination therapy did not cause significant toxicities and was well tolerated. We noted that combination therapy resolved the tEPs faster than MTX alone. We now describe the protocol of a larger single arm trial to estimate the efficacy and side effects of combination gefitinib and MTX to treat stable tEPs with hCG 1000–10 000 IU/L.

**Methods and analysis:** We propose to undertake a single-arm multicentre open label trial (in Edinburgh and Melbourne) and recruit 28 women with tEPs (pretreatment serum hCG 1000–10 000 IU/L). We intend to give a single dose of intramuscular MTX (50 mg/m$^2$) and oral gefitinib (250 mg) daily for 7 days. Our primary outcome is the resolution of EP to non-pregnant hCG levels <15 IU/L without requirement of surgery. Our secondary outcomes are comparison of time to resolution against historical controls given MTX only, and safety and tolerability as determined by clinical/biochemical assessment.

**Ethics and dissemination:** Ethical approval has been obtained from Scotland A Research Ethics Committee (MREC 11/AL/0350), Southern Health Human Research Ethics Committee B (HREC 11180B) and the Mercy Health Human Research Ethics Committee (R12/25).

### ARTICLE SUMMARY

**Article focus**
- Protocol of a study to determine:
- Is combination therapy with MTX and gefitinib effective at resolving tEP?
- Is combination therapy with MTX and gefitinib safe and well tolerated?

**Key messages**
- Tubal ectopic pregnancy (tEPs) with hCG levels <1000 IU/L respond well to treatment with intramuscular MTX.
- tEPs with human chorionic gonadotrophin (hCG) levels >1000 IU/L require multiple hospital visits to resolve with MTX and often require surgery.
- Novel combination therapy of MTX and the oral EGFR antagonist, gefitinib, could reduce the number of hospital visits required to resolve tEPs with hCG levels >1000 IU/L.

**Strengths and limitations of this study**
- This is a phase II exploratory efficacy trial, and will be the 'first in man' to examine the efficacy of gefitinib and MTX to treat tEPs with hCG levels >1000 IU/L
- This is a 'single arm' trial. The data will be used to inform a future large multicentre randomised controlled trial comparing combination therapy to conventional management of tEPs.
- The combination therapy described also has potential use in other pregnancy disorders where medical regression of placental tissue could be useful, for example, molar disease and regression of placenta accrete postpartum.

Data will be presented at international conferences and published in peer-reviewed journals.
**Trial registration number:** ACTRN12611001056987.

## INTRODUCTION

Tubal ectopic pregnancy (tEP) is the most common life-threatening condition in modern gynaecology in both the developed and developing world.[1] [2] tEPs with pre-treatment serum human chorionic gonadotrophin (hCG) levels <1000 IU/L respond well to outpatient medical treatment with an intramuscular injection of methotrexate (MTX). Indeed, it has been suggested that these tEPs could be managed safely, and equally efficiently by expectant management without medical intervention.[3–5] In contrast, single-dose MTX is only cost-effective in women with serum hCG concentrations <1500 IU/L.[6] In tEPs with higher hCG levels (>60% of total tEPs), emergency laparoscopic surgical excision (with its inherent risks of damage to visceral organs) remains the most effective treatment. tEPs with higher hCG levels take a significant time to resolve with MTX and require multiple outpatient monitoring visits. There, therefore, exists a need for more effective medical treatments for tEPs with higher hCG levels to reduce the need for emergency surgery and reduce the time to resolution associated with MTX management.

Gefitinib is an orally active epidermal growth factor receptor (EGFR) antagonist licensed to treat non-small-cell lung cancer.[7] In preclinical studies, we found that EP implantation sites express high levels of EGFR and that gefitinib augments MTX-induced regression of pregnancy-like tissue.[8] To translate this into clinical care, we performed a phase I single-arm open-label dose-escalation study administering a combination of 250 mg oral gefitinib (one dose (n=3), three daily doses (n=3), seven daily doses (n=6)) and intramuscular MTX (50 mg/m$^2$) to 12 women with tEPs.[9] The combination therapy did not cause any significant toxicities, and was well tolerated. We noted that resolution (fall in serum hCG to <15 IU/l) with combination therapy was faster than the median time for tEPs to resolve with MTX alone when compared with contemporaneous controls (21 vs 32 days).

## OBJECTIVES

The objective of this trial is to evaluate the efficacy and side effects of combination gefitinib and MTX to treat tEPs (hCG 1000–10 000 IU/L).

## METHODS AND ANALYSIS
### Study design

Phase II single-arm multicentre open label trial (Edinburgh and two sites in Melbourne).

### Subjects

Twenty-eight women with tEPs with hCG levels 1000–10 000 IU/L.

### Study settings

We intend to recruit patients from gynaecology departments within NHS Lothian (UK), and Southern Health and Mercy Health networks in Melbourne, Australia.

### Sample size

We have calculated the sample size using A'Hern's formula for phase II one-stage designs.[10] For treatment of tEPs with hCG levels 1000–10 000 IU/L by MTX /gefitinib to be considered effective, we expect a success rate of at least 90%.[11] However, a success rate of 70% or less would be considered unacceptable. With 80% power and a 5% level of significance, 28 patients are required to enable us to assess whether the proportion of patients with a successful outcome to treatment is ≤70% or ≥90%. If 24, or more, patients have a successful outcome, we can reject the hypothesis that the true efficacy of MTX /gefitinib is ≤70% and progress to a phase III trial.

### Inclusion criteria

Women aged between 18 and 45 years; pretreatment serum hCG of 1000–10 000 IU/L (rising or static); ultrasound diagnosis of definite tEP (extrauterine gestational sac with yolk sac and/or embryo, with or without cardiac activity) or probable tEP (inhomogeneous adnexal mass or extrauterine sac-like structure)[12] performed by a clinical team of trained, qualified and experienced ultrasonographers; no clinical evidence of intra-abdominal bleeding; no pallor; no guarding/rigidity on abdominal examination; stable blood pressure and heart rate; haemoglobin on full blood examination at day 1 between 100 and 165 g/L).

### Exclusion criteria

Women with a pregnancy of unknown location; evidence of a significant intra-abdominal bleed on ultrasound defined by free fluid above the uterine fundus or the surrounding ovary[13]; women with a history of any significant pulmonary disease; abnormal liver/renal/haematological indices; significant pre-existing dermatological conditions; significant pre-existing gastrointestinal medical illnesses; Japanese ethnicity.

### Participant enrolment

All gynaecology consultants within NHS Lothian (UK), Southern Health and Mercy Health (both Australia) will be sent a letter informing them of the study and requesting permission to approach their patients. The clinical research team in NHS Lothian, Southern Health and Mercy Health will approach eligible women, provide them with patient information sheets and offer them the opportunity to discuss the trial, and obtain informed consent. Consent will only be taken once the patient has had ample time to read the patient information sheet and had her questions answered.

### Intervention

Eligible women will be given a single-dose intramuscular MTX (50 mg/m$^2$) injection with seven daily doses oral gefitinib (250 mg). The gefitinib will be started on the same date when the MTX injection is given.

## Data collection
### Data storage
A log with the patients' name and date of birth will be kept along with their unique study number in a separate file. All the data generated from the study will be stored in an anonymised form in a bespoke database, which will also be password protected. Only anonymised information will be stored on this, and participants will only be identifiable by their study number. All paperwork will be kept in a locked filing cabinet in a locked office. All data will be stored on university server (University of Edinburgh) on a password-protected computer with limited access to the research team, in accordance with the Data Protection Act (UK).

### Screening
A member of the research team will carry out a screening visit to assess eligibility. All data will be recorded on a case record form and transferred to a secure database.

### Participant log
The clinical research team will keep an electronic log of women who fulfil the eligibility criteria, women who are invited to participate in the study, women recruited and women who leave the trial early. Reasons for non-recruitment (eg, non-eligibility, refusal to participate, administrative error) will also be recorded. We will attempt to collect reasons for non-participation from women who decline to take part after previously providing contact details. During the course of the study, we will document reasons for withdrawal from the study and loss to follow-up (figure 1).

### Assessments
To monitor treatment response, we will follow protocols used clinically for medical management with MTX. Serum hCG levels will be measured on days 4, 7 and 11, then weekly until hCG levels drop to non-pregnant levels (<15 IU/L). Medical management will be discontinued and patients will undergo surgery based on their response to MTX and clinical picture (eg, clinical evidence of intra-abdominal bleeding) following standard clinical paradigms documented by the assessing clinician. Participants will be contacted at 3 and 6 months post-treatment to document return of menstrual cycles and any subsequent pregnancies. To monitor safety and tolerability, women will be assessed clinically (history) and biochemically (haematological, renal and liver function tests) on days 4 and 7 (or if elevated, until return to normal physiological levels).

### Primary outcome
Our primary outcome is resolution of tEP without requirement for surgery. Resolution is defined by serum hCG levels (the current clinical marker to monitor treatment response) falling to non-pregnant levels (hCG <15 IU/L). We have selected our primary outcome based on the data from our phase I trial where two

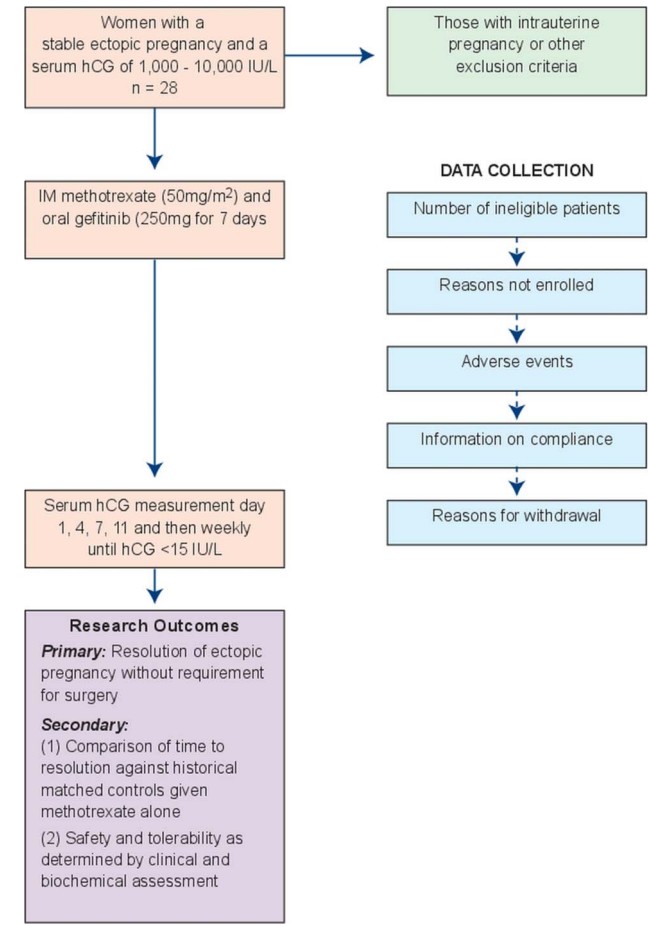

**Figure 1** Flow chart of participants involved in the study.

patients recruited with pretreatment hCG levels >1000 IU/L required surgery and previously published data.[14] We are using a cut-off of <15 IU/L, which corresponds to a negative urinary pregnancy test using the most sensitive assays.

### Secondary outcome
I. Time to resolution (categorical variable) compared with historical controls of similar pretreatment serum hCG levels (identified by an individual blinded to the study).
II. Safety and tolerability as determined by clinical and biochemical assessment. Both MTX and gefitinib have the potential to affect haematological, renal and liver function.

### Proposed analyses
Given this is a single arm efficacy trial, the majority of the data will be expressed as descriptive statistics.

### Ethics and dissemination
Ethical approval has been obtained from the Scotland A Research Ethics Committee (LREC 12/SS/0005) (UK), the Southern Health Human Research Ethics Committee B (SH HREC 11180B) and the Mercy Health Human Research Ethics Committee (R12/25) (both

Australia). Data will be presented at international conferences and published in peer-reviewed journals. We will make the information obtained from the study available to the public through national bodies and charities (eg, Ectopic Pregnancy Trust).

## Adverse events

Participants will collect information about adverse events in their treatment diaries. However, they will be instructed to contact the clinical research team at any time after consenting to join the trial if they have an event that requires hospitalisation or an event that results in persistent or significant disability or incapacity. Any serious adverse events that occur after joining the trial will be reported in detail in the participant's medical notes, followed up until resolution of the event and reported to the ACCORD Research Governance (http://www.accord.ed.ac.uk) and QA Office based at the University of Edinburgh, or the Southern Health/ Mercy Health Human Research Ethics Committees and Therapeutic Goods Administration of Australia's Office of Scientific Evaluation immediately or within 24–72 h.

## DISCUSSION

If effective, we believe that this combination (gefitinib and MTX) could become standard of care for stable tEPs. The combination also has potential use in other pregnancy disorders. There may be other important conditions where medical regression of pregnancy tissue could be useful, for example, women with complete molar pregnancies and persistent molar disease and women with placenta accrete postpartum (to avoid hysterectomy).

Regarding the safety of gefitinib, data from post marketing surveillance representing over 92 000 patients exist and have shown that EGFR inhibitors are well tolerated and largely free of serious side effects (Food and Drug Administration (FDA) report).[15] Of note, the data on tolerability are based on patients taking gefitinib daily on an ongoing, indefinite basis, after primary treatment of cancer. Diarrhoea and skin rash are the most common side effects (20–30%). The skin rash, described as acneiform, can be severe, but is generally self-limited. Skin rashes occur within a month of initiation of treatment, but rarely in the first week. Interstitial lung disease (ILD) is a very rare but a serious side effect of gefitinib. It is a thickening of the lung parenchyma that can be fatal in a third of cases. Of the 31 045 patients in the USA who took gefitinib (reported to the FDA), 84 developed ILD (0.3%). We plan to administer seven 250 mg gefitinib tablets, one daily for only 7 days, in addition to MTX. This is an extremely short duration of treatment compared with gefitinib's current marketing indications and existing data usage. We would not expect this short course to have an adverse long-term effect on fertility but we will be assessing participants 3 and 6 months posttreatment to document return of menstrual cycles and any subsequent pregnancies.

We do not anticipate that this will be the final trial to determine whether further exploration of combination therapy with gefitinib and MTX is worthwhile. We hope that the study will generate sufficient 'signal' that gefitinib and MTX may be effective and safe, to support a funding application for a larger trial with a comparative group. Such a trial could be designed as an 'equivalence' trial in terms of treatment efficacy between conventional management and the gefitinib/MTX comparison. It would aim to test the hypothesis that gefitinib/MTX was superior in a range of outcomes prioritised by consumer groups and clinicians. We anticipate that these outcomes could include: time to resumption of normal activities, SF-36 at intervals after treatment and patient satisfaction scores. Outcomes of a subsequent pregnancy are also important but would require long-term follow-up studies. We anticipate that focus groups and surveys of patients and clinicians would be required to define the outcomes (other than efficacy) of these studies.

**Author affiliations**
[1]MRC Centre for Reproductive Health, University of Edinburgh, Edinburgh, UK
[2]Translational Obstetrics Group, University of Melbourne, Mercy Hospital for Women, Melbourne, Australia
[3]Monash Institute of Medical Research, Clayton, Australia
[4]Obstetrics and Gynaecology, Division of Applied Clinical Sciences, University of Aberdeen, Aberdeen Maternity Hospital, Aberdeen, UK
[5]Mercy Hospital for Women, Melbourne, Australia

**Contributors** AH and ST were involved in research, contribution of original material, editing and approval of final manuscript; AD and MS were also involved in contribution of original material, editing and approval of final manuscript; MS, MR, CD, EW, HC, TJ, SB, JM and JN edited and approved the final manuscript.

**Funding** This work is supported by an NHMRC Grant (#1008276) to ST, TJ and EW, and an MRC Centenary Award (G0802808) to AH. The funders will have no role in the study design; collection, management, analysis and interpretation of data; writing of the report; and the decision to submit the report for publication.

**Competing interests** AH is funded by an MRC Clinician Scientist Fellowship and MRC Centenary Award (G0802808) and holds grants from the Chief Scientist's Office Scotland (CZH/4/688) (HC coinvestigator) and Well-being of Women. HC holds an MRC DCS Grant (G003611), an MRC Centre Grant (G1002033) and research collaboration funding from Bayer Pharma AG. UN, TJ and ST are joint holders of patents that relate to the use of EGFR inhibition in treating ectopic pregnancies.

**Ethics approval** Ethical approval has been obtained from Scotland A Research Ethics Committee (MREC 11/AL/0350), Southern Health Human Research Ethics Committee B (HREC 11180B) and the Mercy Health Human Research Ethics Committee (R12/25).

**Provenance and peer review** Not commissioned; externally peer reviewed.

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
