## [Reviewer comments · BMJ Open]

Some articles will have been accepted based in part or entirely on reviews undertaken for other BMJ Group journals. These will be reproduced where possible.

ARTICLE DETAILS

TITLE (PROVISIONAL)	Phase II single arm open label multicentre clinical trial to evaluate the efficacy and side effects of combination gefitinib and methotrexate to treat ectopic pregnancies (GEM II): study protocol
AUTHORS	Horne, Andrew; Skubisz, Monika; Doust, Ann; Duncan, W; Wallace, Euan; Critchley, Hilary; Johns, Terry; Norman, Jane; Bhattacharya, Siladitya; Mollison, Jill; Rasmussen, Michael; Tong, Stephen

VERSION 1 - REVIEW

REVIEWER	Dr Dimitrios Mavrelou Clinical Academic Lecturer in Reproductive Medicine University College London
REVIEW RETURNED	05-Apr-2013

GENERAL COMMENTS	The diagnostic criteria for ultrasound diagnosis of ectopic pregnancy are unclear. In particular "abnormal mass" should be defined. The ultrasound diagnostic method should be detailed and specific diagnostic criteria should be defined. The inclusion/exclusion criteria do not clarify whether the morphology of the ectopic will be used to exclude women. Specifically, do the authors plan to exclude women with a live ectopic pregnancy? The inclusion criteria do not specify whether an upper limit of hCG will be used. This should be made explicit. In order to avoid ambiguity it may be useful to specify that women with pregnancy of unknown location will be excluded from the study. The methods do not specify criteria for intervention i.e. surgery. The authors should clarify their indications for discontinuation of current management. The strengths and limitations section of the study proposal mentions the use of historical controls. The sample size section of the manuscript does not mention how the controls will be selected. There is a discrepancy between the primary outcome as mentioned in the manuscript (resolution within 25 days) and the figure (resolution within 21 days).
--

REVIEWER	Davor Jurkovic, Consultant Gynaecologist University College Hospital London UK
-----------------	--

	Competing interests: none
REVIEW RETURNED	09-Apr-2013

GENERAL COMMENTS	This is an interesting project which aims to assess the value of a novel approach to medical treatment of ectopic pregnancy. I have the following comments to make: Page 6, line 5 The authors should make it clear that they are planning to use this new treatment to manage tubal ectopic pregnancies as ectopic pregnancies in other locations often follow a very different clinical course which could affect the result of this small trial. Page 6, line 11 The authors refer here and throughout the text to ectopic pregnancies presenting with serum hCG ≥ 1500 IU/l as 'large'. This is inappropriate as the size of ectopic correlates poorly with the serum hCG levels. For the purpose of this trial it would be better if they remained consistent by dividing all ectopics in two groups: those presenting with low initial hCG (< 1500 IU/l) and high initial hCG (≥ 1500 IU/l). Page 6, line 11 The authors are correct in pointing out that approximately 60-70% of women presenting with tubal ectopics are currently treated surgically. Majority of them, however, present with severe symptoms or with signs of rupture. In view of that this new medical treatment would be potentially helpful in no more than 30% of ectopics presenting to acute gynaecological units. Page 7, line 2 Side effects of gefitinib are very common with 75% of patients experiencing skin changes and 50% diarrhoea. The drug may also have an adverse effect on women's fertility (http://www.cancerresearchuk.org/cancer-help/about-cancer/treatment/cancer-drugs/gefitinib). The manufacturer's website (http://www.gefitinib.org/sideeffects.html) states that the drug may also impair clotting and increase the risk of bleeding, which is of particular concern if it was to be used to treat ectopic pregnancy. Side effects of methotrexate have also been well-documented in the past. The author's initial experience with a cohort of 12 patients does not provide reassurance that side-effects with this particular therapeutic approach are unlikely to occur. I would suggest that the discussion is expanded to elaborate more on these issues. Page 7, line 4 The authors refer to their previous study which provided them with some information regarding the potential value of gefitinib in the treatment of tubal ectopic pregnancy. They state that with the combination therapy the median time for ectopics to resolve was faster compared with methotrexate alone (21 versus 32 days). Was this result statistically significant? Was it obtained in women presenting with high initial hCG levels or did they also include women with low serum hCG as well? Page 8, line 15 Sample size calculation is difficult to understand, but I presume that the authors had obtained an expert statistical advice. I do fear, however, that the presumed 95% efficacy rate of the combination treatment is unrealistically high. Page 9, line 1 Ultrasound diagnosis of an abnormal mass outside the uterine cavity is not specific enough to diagnose a tubal ectopic
---

	pregnancy as it would allow misclassification of any incidental ovarian or tubal abnormalities as an ectopic pregnancy. It is also of critical importance that the authors provide a very clear definition of what would be considered a significant intra-abdominal bleeding on ultrasound scan. I am concerned that the presence of embryonic cardiac activity, actual size of ectopic and size of haematosalpinx are not taken into consideration when selecting women for medical treatment. For example, that would allow for a woman with a live eight weeks' size ectopic and initial hCG of 25000 IU/l to be included into the study. In a case like that the risk of serious maternal morbidity with medical management is very high and the authors should consider re-defining their selection criteria. Page 9, line 14 Women will be recruited from wide geographical areas. It is inevitable that some of them will have serial measurements of serum hCG taken before the diagnosis is reached. Would that affect inclusion into the trial? What measures will be put in place to check the quality and accuracy of ultrasound diagnosis before the treatment is commenced? Page 11, line 10 The criteria for stopping medical management such as suspected rupture, non-declined of hCG during early follow up, etc. should also be described. Page 12, line 5 The authors have chosen to define the success of medical management as a drop of serum hCG to <5IU/l 25 days after initiation of treatment. This would mean that a clinically stable woman with declining hCG of 10IU/l at day 25 would be classified as failing medical management, which is at odds with standard clinical practice. Page 14, line 8 I cannot see many clinicians wishing to use chemotherapy to treat women with intrauterine retained products of conception or for termination of an intrauterine pregnancy.
--	--

REVIEWER	Dr Dharani Hapangama Clinical Senior Lecturer / Consultant in Obstetrics & Gynaecology Department of Women's and Children's Health Institute of Translational Medicine University Department Liverpool Women's Hospital Crown Street Liverpool L8 7SS
REVIEW RETURNED	10-Apr-2013

THE STUDY	This interesting study protocol is aimed to examine the therapeutic potential of combination medical treatment for ectopic pregnancies. The inclusion of a patient group, particularly women with larger ectopics who currently have no alternative but surgical treatment and the subsequent loss of the affected tube, is of great clinical significance. Hence, the results of this proposed study has an important and novel clinical implication for the management of ectopic pregnancies. The relevant ethical committees have already granted approval for the study, yet the follow up arrangement of patients remain to be one of my concerns. The very fact that researchers intend to include
------------------	---

	a group of women with larger ectopic pregnancies who are currently offered surgical treatment fairly immediately (usually within 24 h) and are not offered medical treatment because of their presumed increased risk of rupture / catastrophic bleeding remains a worry. The CMACE report "Saving mothers lives" has highlighted the contribution of ectopic pregnancies in to the maternal deaths in the UK and this has been falling seemingly due to the early detection and appropriate surgical treatment. Still around 0.2 out of 100 ectopic pregnancies result in death, many due to substandard care and late surgical intervention. What are the extra precautions the authors will take in the very likely case of rupturing of these larger ectopic pregnancies whilst on the proposed medical treatment? Will those women with a diagnosis of larger ectopic who consent to try the proposed treatment be observed as inpatients? It may be useful if the protocol elaborate on this aspect in particular. I only have few other minor queries, mentioned below;  1. Inclusion criteria; are all these criteria need to be met? What about women with >1500IU/L hCG without USS evidence of an ectopic mass? 2. What is the defined 'normal' haemoglobin level for the purpose of this study? 3. Typos; Page 9 L2, repeated word
--	--

VERSION 1 – AUTHOR RESPONSE

REVIEWER 1

QUESTION: The diagnostic criteria for ultrasound diagnosis of ectopic pregnancy are unclear. In particular "abnormal mass" should be defined. The ultrasound diagnostic method should be detailed and specific diagnostic criteria should be defined.

RESPONSE: We have made the criteria clearer (see page 9 lines 3-7) and based them on the 2011 consensus statement on 'definitions' of pregnancies of unknown location (Fertil Steril 2011 95(3):857-66).

QUESTION: The inclusion/exclusion criteria do not clarify whether the morphology of the ectopic will be used to exclude women. Specifically, do the authors plan to exclude women with a live ectopic pregnancy?

RESPONSE: See above. We do not plan to exclude live ectopic pregnancies.

QUESTION: The inclusion criteria do not specify whether an upper limit of hCG will be used. This should be made explicit.

RESPONSE: This has been included (see page 9 lines 2-3). Our upper limit is 10,000IU/L.

QUESTION: In order to avoid ambiguity it may be useful to specify that women with pregnancy of unknown location will be excluded from the study.

RESPONSE: We have now specified this in the exclusion criteria (see page 9 line 13)

QUESTION: The methods do not specify criteria for intervention i.e. surgery. The authors should clarify their indications for discontinuation of current management.

RESPONSE: We have included criteria in the 'Assessments' section (see page 11 lines 19-22).

QUESTION: The strengths and limitations section of the study proposal mentions the use of historical controls. The sample size section of the manuscript does not mention how the controls will be selected.

RESPONSE: Historical controls of similar baseline serum hCG levels (hCG level on day of treatment) for each case will be identified by an individual blinded to the study. This statement has been included in the section on 'Secondary Outcome' (page 12 lines 15-17).

QUESTION: There is a discrepancy between the primary outcome as mentioned in the manuscript (resolution within 25 days) and the figure (resolution within 21 days).

RESPONSE: This has been amended. Please note the primary outcome is now changed in response to reviewer 2 (see below).

REVIEWER 2

QUESTION: Page 6, line 5 The authors should make it clear that they are planning to use this new treatment to manage tubal ectopic pregnancies as ectopic pregnancies in other locations often follow a very different clinical course which could affect the result of this small trial.

RESPONSE: We have tried to make this clearer with changes throughout the manuscript, referring to the ectopic pregnancies as 'tubal ectopic pregnancy' or 'tEP'.

QUESTION: Page 6, line 11 The authors refer here and throughout the text to ectopic pregnancies presenting with serum hCG ≥ 1500 IU/l as 'large'. This is inappropriate as the size of ectopic correlates poorly with the serum hCG levels. For the purpose of this trial it would be better if they remained consistent by dividing all ectopics in two groups: those presenting with low initial hCG (< 1500 IU/l) and high initial hCG (≥ 1500 IU/l).

RESPONSE: This has been changed throughout the manuscript. Please note that the group of patients that we plan to study has been amended to women with hCG 1,000-10,000IU/L.

QUESTION: Page 6, line 11 The authors are correct in pointing out that approximately 60-70% of women presenting with tubal ectopics are currently treated surgically. Majority of them, however, present with severe symptoms or with signs of rupture. In view of that this new medical treatment would be potentially helpful in no more than 30% of ectopics presenting to acute gynaecological units.

RESPONSE: This is a good point although in our units severe symptoms, or signs of rupture, requiring surgery are present in $< 50\%$ of patients.

QUESTION: Page 7, line 2 Side effects of gefitinib are very common with 75% of patients experiencing skin changes and 50% diarrhoea. The drug may also have an adverse effect on women's fertility (<http://www.cancerresearchuk.org/cancer-help/about-cancer/treatment/cancer-drugs/gefitinib>). The manufacturer's website (<http://www.gefitinib.org/sideeffects.html>) states that the drug may also impair clotting and increase the risk of bleeding, which is of particular concern if it was to be used to treat ectopic pregnancy. Side effects of methotrexate have also been well documented in the past. The author's initial experience with a cohort of 12 patients does not provide reassurance that side effects with this particular therapeutic approach are unlikely to occur. I would suggest that the discussion is expanded to elaborate more on these issues.

RESPONSE: We acknowledge the reviewer's concerns. Data from post marketing surveillance

representing over 92,000 patients exists, and has shown that EGFR inhibitors are well tolerated and largely free of serious side effects (FDA report: Clin Cancer Res 2004;10:1212-8) . Of note, the data on tolerability is based on patients taking gefitinib daily on an ongoing, indefinite basis, after primary treatment of a cancer. Diarrhea and skin rash are the most common side effects (20-30%). The skin rash, described as acneiform, can be severe, but is generally self-limited. Skin rashes occur within a month of initiation of treatment, but rarely in the first week. Interstitial lung disease (ILD) is a very rare but serious side effect of gefitinib. It is a thickening of the lung parenchyma that can be fatal in a third of cases. Of the 31,045 patients in the USA who took gefitinib (reported to the FDA), 84 developed ILD (0.3%). We plan to administer seven 250mg gefitinib tablets, one daily for only seven days, in addition to methotrexate. This is an extremely short duration of treatment compared with gefitinib's current marketing indications and existing data usage. We would not expect this short course to have an adverse long-term effect on fertility but we will be assessing participants 3 and 6 months post treatment to document return of menstrual cycles and any subsequent pregnancies. We have included this in our discussion (see page 14, lines 10-25, page 15 lines 1-2).

QUESTION: Page 7, line 4 The authors refer to their previous study which provided them with some information regarding the potential value of gefitinib in the treatment of tubal ectopic pregnancy. They state that with the combination therapy the median time for ectopics to resolve was faster compared with methotrexate alone (21 versus 32 days). Was this result statistically significant? Was it obtained in women presenting with high initial hCG levels or did they also include women with low serum hCG as well?

RESPONSE: Of six participants treated with combination gefitinib and methotrexate, with a pre-treatment human chorionic gonadotrophin (hCG) between 1000-3000IU/L, median hCG levels more than halved by days 4 and 7, and declined significantly faster than 71 contemporaneous women treated with methotrexate alone. The median time for the ectopic pregnancies to resolve with combination therapy was 34% shorter compared to methotrexate (21 vs 32 days; $p=0.018$).

QUESTION: Page 8, line 15 Sample size calculation is difficult to understand, but I presume that the authors had obtained an expert statistical advice. I do fear, however, that the presumed 95% efficacy rate of the combination treatment is unrealistically high.

RESPONSE: Expert statistical advice was provided for the study, based on our initial observations, but we have taken this comment on board. Please note the changes to the sample size calculation due to this comment (we hope that this section is now clearer) and due to the change in our primary outcome (see below).

QUESTION: Page 9, line 1 Ultrasound diagnosis of an abnormal mass outside the uterine cavity is not specific enough to diagnose a tubal ectopic pregnancy as it would allow misclassification of any incidental ovarian or tubal abnormalities as an ectopic pregnancy. It is also of critical importance that the authors provide a very clear definition of what would be considered a significant intra-abdominal bleeding on ultrasound scan. I am concerned that the presence of embryonic cardiac activity, actual size of ectopic and size of haematosalpinx are not taken into consideration when selecting women for medical treatment. For example, that would allow for a woman with a live eight weeks' size ectopic and initial hCG of 25000 IU/l to be included into the study. In a case like that the risk of serious maternal morbidity with medical management is very high and the authors should consider re-defining their selection criteria.

RESPONSE: We hope that we have made the criteria clearer (see page 9 lines 3-7) and based them on the 2011 consensus statement on 'definitions' of pregnancies of unknown location (Fertil Steril 2011 95(3):857-66). We also have included a 'ceiling' pre-treatment hCG level of 10,000IU/L. We have defined a 'significant intra-abdominal bleed' on ultrasound scan as 'free fluid above the

umbilicus or surrounding the ovary' (see World J Emerg Surg 2007 2:23).

QUESTION: Page 9, line 14 Women will be recruited from wide geographical areas. It is inevitable than some of them will have serial measurements of serum hCG taken before the diagnosis is reached. Would that affect inclusion into the trial? What measures will be put in place to check the quality and accuracy of ultrasound diagnosis before the treatment is commenced?

RESPONSE: For inclusion into the trial, pre-treatment hCG levels will be required to be rising or static (see page 9 line 3). All of the ultrasound assessments will be performed by a clinical team of trained, qualified and experienced ultrasonographers.

QUESTION: Page 11, line 10 The criteria for stopping medical management such as suspected rupture, non-declined of hCG during early follow up, etc. should also be described.

RESPONSE: See response to reviewer 1.

QUESTION: Page 12, line 5 The authors have chosen to define the success of medical management as a drop of serum hCG to <5IU/l 25 days after initiation of treatment. This would mean that a clinically stable woman with declining hCG of 10IU/l at day 25 would be classified as failing medical management, which is at odds with standard clinical practice.

RESPONSE: We thank the reviewer for this important comment. We have changed our primary outcome to resolution of ectopic pregnancy without requirement for surgery. We have also adjusted the definition of 'resolution' to <15IU/L, which corresponds to a negative urinary pregnancy test using the most sensitive assays (see page 12 lines 4-12).

QUESTION: Page 14, line 8 I cannot see many clinicians wishing to use chemotherapy to treat women with intrauterine retained products of conception or for termination of an intrauterine pregnancy.

RESPONSE: We have removed this from the discussion.

REVIEWER 3

QUESTION: The relevant ethical committees have already granted approval for the study, yet the follow up arrangement of patients remain to be one of my concerns. The very fact that researchers intend to include a group of women with larger ectopic pregnancies who are currently offered surgical treatment fairly immediately (usually within 24 h) and are not offered medical treatment because of their presumed increased risk of rupture / catastrophic bleeding remains a worry. The CMACE report "Saving mothers lives" has highlighted the contribution of ectopic pregnancies in to the maternal deaths in the UK and this has been falling seemingly due to the early detection and appropriate surgical treatment. Still around 0.2 out of 100 ectopic pregnancies result in death, many due to substandard care and late surgical intervention. What are the extra precautions the authors will take in the very likely case of rupturing of these larger ectopic pregnancies whilst on the proposed medical treatment? Will those women with a diagnosis of larger ectopic who consent to try the proposed treatment be observed as inpatients? It may be useful if the protocol elaborate on this aspect in particular.

RESPONSE: This is an important point. We are aware that a major contributing factor to the maternal deaths was lack of awareness of patients regarding the diagnosis of ectopic pregnancy. All patients undergoing medical management (whether it is methotrexate alone, or methotrexate with gefitinib) in our units are carefully informed of the symptoms that indicate possible rupture. They have emergency

numbers to contact, and all staff follow defined care pathways in these situations. In addition, we hope that we alleviate the reviewer's concerns by the following clarifications and additions within the protocol: (1) We have tried to make the criteria for inclusion clearer (see page 9 lines 1-10) and based them on the 2011 consensus statement on 'definitions' of pregnancies of unknown location (Fertil Steril 2011 95(3):857-66); (2) We also have included a 'ceiling' pre-treatment hCG level of 10,000IU/L; (3) We have also defined a 'significant intra-abdominal bleed' on ultrasound scan as 'free fluid above the umbilicus or surrounding the ovary' (see World J Emerg Surg 2007 2:23).

QUESTION: Inclusion criteria; are all these criteria need to be met? What about women with >1500IU/L hCG without USS evidence of an ectopic mass?

RESPONSE: We will not include pregnancies of unknown location (see page 9 line 13).

QUESTION: What is the defined 'normal' haemoglobin level for the purpose of this study?

RESPONSE: We have added a definition (see page 9 lines 9-10)

QUESTION: Typos; Page 9 L2, repeated word

RESPONSE: This has been corrected.

VERSION 2 – REVIEW

REVIEWER	Davor Jurkovic Consultant Gynaecologist University College Hospital London, UK
REVIEW RETURNED	04-Jun-2013

- The reviewer completed the checklist but made no further comments.